# Protein Ingredients in Bread: Technological, Textural and Health Implications

**DOI:** 10.3390/foods11162399

**Published:** 2022-08-10

**Authors:** Pavel Prieto-Vázquez del Mercado, Luis Mojica, Norma Morales-Hernández

**Affiliations:** Tecnología Alimentaria, Centro de Investigación y Asistencia en Tecnología y Diseño del Estado de Jalisco, A.C. Unidad Zapopan, Zapopan 45019, Mexico

**Keywords:** protein sources, nutritional improvement, texture, bread

## Abstract

The current lifestyle and trend for healthier foods has generated a growing consumer interest in acquiring bread products with a better nutritional composition, primarily products with high protein and fiber and low fat. Incorporating different protein sources as functional ingredients has improved the nutritional profile but may also affect the dough properties and final characteristics of bread. This review focuses on the incorporation of different animal, vegetable, and mixed protein sources, and the percentage of protein addition, analyzing nutritional changes and their impact on dough properties and different texture parameters, appearances, and their impact on bread flavor and health-related effects. Alternative processing technologies such as germination and sourdough-based technologies are discussed. Using fermented doughs can improve the nutritional composition and properties of the dough, impacting positively the texture, appearance, flavor, and aroma of bread. It is essential to innovate alternative protein sources in combination with technological strategies that allow better incorporation of these ingredients, not only to improve the nutritional profile but also to maintain the texture and enhance the sensory properties of the bread and consequently, increase the effects on consumer health.

## 1. Introduction

Wheat-based products such as bakery products are one of the primary sources of daily energy intake. They are an important source of carbohydrates, dietary fiber, micronutrients (vitamins and minerals), and antioxidants [1]. *Triticum aestivum* L. and *Triticum durum* are the most commonly used wheat species, where 95% of wheat production corresponds to *Triticum aestivum* L. or common wheat principally used for breadmaking [2].

The primary component of wheat flour is starch, followed by proteins. About 80% of protein content is represented by the gluten-forming protein, a complex mixture of gliadins and glutenins. The gluten matrix confers the properties to form a cohesive dough with the extensibility and elasticity properties to allow the growth of bubbles and gas retention necessary to obtain a porous product [3]. The flour is kneaded into a dough, adding water and yeast, generating a leveled product, such as bread [4].

Consumers are interested in healthier food products that prevent nutrition-related diseases and improve physical and mental well-being [5]. Food products such as bread could act as vehicles incorporating functional ingredients generating health benefits for consumers [6,7].

Enhancing nutrition in bread is an interesting opportunity for the food industry. Most breads are made with refined wheat flour because it generates a high loaf volume, light color, homogeneous crumb porosity, and soft crumb. These breads are widely accepted by customers, although they lack vitamins, minerals, lysine, dietary fiber, and antioxidant components, and present a high glycemic index [2,8,9]. This deficiency could be improved with whole wheat flour, but consumers prefer refined wheat bread [2] or other alternatives for better nutritional balance.

The trend is to use alternative protein sources in refined wheat bread to enhance protein and fiber content and improve amino acid balance. Also, these ingredients could increase the antioxidant potential and reduce the glycemic index in bread [8,10]. Animal and vegetable ingredients such as milk derivates, edible insects, fish derivates, legumes, cereals, pseudocereals, and other sources (Figure 1) are commonly used in different ways, including flours, powders, protein concentrates, protein isolates and protein hydrolysates. These protein sources could improve nutritional content and offer health benefits.

However, other parameters such as dough rheology, texture, and other sensory characteristics, including appearance, flavor, and taste are modified compared with traditional bread. The lack of gluten in the different protein sources interferes with the final product quality. Some authors prefer to use high protein ingredients (80–90%) in small amounts compared to whole flour, reaching the same protein content and lowering the wheat flour reduction [11].

The bread industry should focus its attention on offering healthier products with high protein content, in addition to products with sensory characteristics attractive to consumers. The objective of this review was to provide current information on the enrichment of refined wheat bread with protein ingredients from different sources and how it affects the dough behavior and technological properties of the bread matrix, in addition to the impact of novel ingredients on sensory characteristics and the biologic potential of nutritionally improved bread.

## 2. Impact of Protein Addition on the Nutritional Composition of Wheat Bread

Protein incorporation in bread could provide healthy alternatives and health benefits to consumers [6]. Adding protein from different sources could affect the nutritional, chemical, physical, and functional properties of bread. Different authors use various sources and levels of protein to explore their effects on the sensory and nutritional characteristics of bread.

Protein ingredients can be obtained from vegetable or animal sources in different forms, including flours or powders, protein concentrates, protein isolates, protein hydrolysates, or fresh ingredients, whose protein content differs from each other (flours: protein < 65%, protein concentrates: protein > 65% and protein isolates: protein > 90%) [12]. In addition, the nutritional quality of protein differs between sources by the obtaining process, composition of essential amino acids, its digestibility, bioavailability, and the presence of antinutritional factors [13].

Table 1 depicts the nutritional composition of different groups of protein-rich ingredients, where most of them are processed as flours or with some additional steps.

Legumes and pseudocereal-derived ingredients are commonly used in bread products due to their water-holding capacity, solubility, emulsifying, foaming, and gelling properties. They are a good source of high-quality carbohydrates, fiber, protein, and micronutrients. The amino acid profile in legumes contains a low level of sulfur amino acids but a high level of lysine, which could be an excellent complement to the amino acid profile of wheat flour, which lacks lysine [29]. Similarly, pseudocereals’ amino acid profile complements lysine-lacking wheat flour. Moreover, pseudocereal and legume ingredients are rich in starch, fiber, micronutrients, and phytochemicals with potential health benefits [30].

Conversely, animal sources such as whey and casein proteins are used in bread. Likewise, they have good technofunctional properties including water-holding capacity, solubility, and gelling properties. Furthermore, they present an excellent amino acid profile rich in lysine, methionine, and tryptophan [31].

The interest in using new protein alternatives, such as insects, has motivated the European Food Safety Authority (EFSA) to propose a list of edible insect species with great potential to be used for food and feed. The amino acid profile in insects is comparable to the meat with lower environmental affection [32,33]. Depending on the insect species used, the composition of protein, fat, micronutrients, and fiber will differ [20,32,34].

European regulation indicates a claim that a food is a “source of protein” when at least 12% of the energy value is provided by protein and a claim “high protein” when at least 20% of the energy value is provided by protein [35]. This regulation is important for customers’ decisions. In Table 1, it can be observed that the bread products studied by all authors could be considered as “source of protein.” However, understanding the impact of texture and sensory parameters by adding high-protein ingredients to bread could make it difficult to formulate good “high-protein” bread products.

Powder protein sources with a high protein content, such as some insects or other animal sources, are good options for increasing protein content with small additions [19,20,22,24]. Protein concentrates or protein isolates from vegetable sources whose protein composition is higher than 65% are an excellent alternative to attain higher protein composition in bread compared to flours [11,17].

Commonly, most vegetable and insect protein sources have an important fiber content. The Food and Drug Administration normatives claim a product as “good source of fiber” when it contains more than 10% of the Daily Reference Value (28 g per day) [36]. Increasing the fiber content in bread could aid in higher daily fiber intake and may have protective effects against cardiovascular disease, diabetes, metabolic syndrome, inflammatory bowel syndrome, diverticular disease, obesity, and some cancer types [37].

Fiber can also provide colon health benefits by acting as a prebiotic for the gut microbiome [38]. Fiber also could prolong satiety after meals [37]. In Table 1, adding plant flours such as hemp [18] and some amaranth species [17] was an excellent alternative to increase fiber content. In the case of tilapia-waste bread, as the incorporation increases, the fiber content decreases and the moisture increases compared to control [24]. Incorporating vegetable and insect sources is a good alternative to improve protein and fiber content.

Products with a high content of protein or fiber are of interest to consumers. Therefore, the bread industry should focus on claims for nutritional improvement without losing aspects such as the quality of dough, good texture in bread, and sensory acceptability.

## 3. Effects of Ingredient Substitution on Dough Quality

Dough is a viscoelastic system structurally composed of starch granules, water-soluble and insoluble proteins, and entrapped air. The cohesiveness and structure are generated by gluten proteins, the main water-insoluble fibrous proteins. Glutenin (elastic properties) and gliadin (viscous properties) form the viscoelastic protein network [39,40].

Before mixing, gluten proteins are tightly coiled, where numerous disulfide bonds are responsible for their rigid structure. To break these disulfide bonds and promote dough expansion, the gluten protein must be mechanically disrupted by mixing the hydrated dough. These broken bonds form sulfhydryl (-SH) groups, which must be oxidized to form new disulfide bonds to fix the new structure of the dough. However, adding ingredients could damage this gluten network by interfering with forming these bonds [39,40,41].

Dough resistance to deformation against mixing blades at a constant temperature and speed could be measured by a farinograph or a mixograph. These instruments help to measure water absorption (WA), mixing tolerance index, degree of softening (DSF), arrival time, dough development time (DDT) and dough stability time (DST). These measurements could be used to predict the baking performance, which is associated with dough extension and gluten network formation [11,31,40].

WA refers to the quantity of water added up to reach maximum dough consistency at the center of the farinogram band [500 Brabender Units (BU)] or could be measured using a mixograph. WA is important in the hydration and the development of the gluten network, affecting bread quality [14,31]. Table 2 depicts a tendency where most vegetable sources increase the water absorption parameter, in contrast to animal sources, including milk proteins and some insects. Zhou et al. (2018) report how whey protein incorporation inhibits the hydration of granular starch and wheat protein. Conversely, soy flour presents high WA capacity generating different dough behaviors compared to whey protein. The nutrient composition of ingredients, the amino acid composition of protein, and the percentage of substitution have an important effect on dough WA [42].

Cappelli et al. (2020) observed a decrease in WA in mealworm dough and an increasing trend in cricket powder, related to the bromatological composition of the powder and the amino acid composition of insect protein. In the case of chickpea flour, there was an increase in WA compared to protein content and a rise in pentosans, particularly ribose and deoxyribose [34]. Pretreatments of protein sources such as germination or toasting before dough formation could impact the dough WA. In the case of germination, the WA was reduced, but the toasting process increased it. This could occur due to starch loss during germination and the protein structure changes. In the toasting process, protein denaturation and starch gelatinization may result in an increase in WA compared with raw flour treatment, but not higher than the dough without addition [14].

Dough development time (DDT) represents the time (minutes) to the maximum development of the dough (optimal viscosity and elasticity properties for gas retention). The mixometer measures the time needed to attain a torque of 1.1 Nm. Conversely, the a farinograph measures the time to the nearest half min from the first addition of water to develop the maximum dough consistency [11,31,41].

There is a tendency for vegetable sources to present higher values of DDT compared with unblended dough, but it still unclear. These higher values could be attributed to the presence of starch, the different physicochemical properties between the components of flours and the higher rate of water absorption caused by soluble proteins of the ingredient used [43]. In the case of animal source doughs, they generate shorter values of DDT. Table 2 depicts how DDT can increase or decrease depending on the protein source used and the level of addition or substitution.

Dough stability time (DDT) in minutes indicates the dough’s resistance to mixing or forming a stable gluten network, whereas lower values indicate softer dough [11,14,41]. The dough stability is commonly related to gluten dilution and the altered water behavior is caused by adding different ingredients [41]. Table 2 indicates how different protein sources could impact dough stability.

Pea flour dough demonstrates lower stability compared to wheat flour. This could be due to the interruption of the starch–protein matrix with the addition of pea flour, generating a weakening of the dough during mixing and a decrease in elasticity [14]. Bread blends with legume protein tend to have higher water absorption, diluting the wheat gluten and weakening the dough strength and stability, altering bread texture properties [44].

There is a relationship between high resistance to dough extension and baking performance. To measure this, a texture analyzer or an extensograph is required, where the force (Kg) needed to break the dough is referred as dough extensibility (EX) [31].

There is a tendency to indicate that animal proteins in bread provide lower EX values compared to vegetable proteins. Gani et al. (2015) suggest that the significant decrease in dough extensibility when adding dairy protein could be due to dilution of gluten content and interaction of whey protein with gluten. This weakening could be due to the interference of milk proteins with sulfhydryl groups during wheat flour dough [31]. Conversely, Graça et al. (2019) added yogurt and curd cheese to bread, where the addition of yogurt positively impacted EX and deformation energy. This could occur by the presence of exopolysaccharides, which act as lubricants together with gliadins. This phenomenon did not happen with curd cheese addition due to competition for available water between proteins [25].

Alternatively, the alveographic method indicates a different result of extensibility (L). It measures the average abscissa at bubble rupture and provides a value for tenacity (P), representing the maximum overpressure needed to blow the dough bubble. In addition, the ratio between P and L (P/L) is important in the technological success of leavened products [20,34].

Osimani et al. (2018) related how different values in tenacity and extensibility in cricket powder blends are the result of the reduced gluten due to difficulties in gluten-network formation during the mixing phase, where P/L ratio demonstrates the highest value with the maximum cricket blend (6.22) and the wheat dough demonstrates the lowest (0.94) [19].

The use of chickpea flour has less impact on L than on insect flour due to different bromatological and amino acid compositions of flours. In contrast, chickpea has a similar bromatological composition compared to wheat. In comparison, chickpea dough demonstrates the lowest P at the maximum blend, related to the presence of enzymes that could disrupt gluten-forming proteins [34].

**Table 2 foods-11-02399-t002:** Dough mixing properties.

Protein Source	Type	Percentage of Addition (%) vs. Water Absorption	Results	Reference
Lupine	Debittered Flour (DLF)	10↑, 15↓, 20↓	↑ DDT in DLF at 10%, ↓ DDT in DLF at 15, 20% and FLF (AL) vs. control. ↑ DS in DLF (AL) and FLF at 10%, ↓ DS in FLF at 15 and 20% vs. control.	[15]
	Fermented Flour (FLF)	10↓, 15↑, 20↓		
Soy (SP)	Protein concentrate	5↑, 10↑, 15↑	↑ DDT and ↑ DS in SP (AL) and in PP at 5% vs. control, but ↓ at 10 and 15% PP vs. control. ↑ DS in PP (AL) vs. control. ↑ Weakening of gluten network in PP at 15%, and SP at 10 and 15%.	[11]
Pea (PP)	Protein concentrate	5↑, 10↑, 15↑		
Pea	Flour (PF)	30↓	Similar DDT in GRF and ↑ DDT in PF and TF vs. control. ↓ DS and ↑ weakening in all treatments vs. control.	[14]
Pea	Germinated Flour (GRF)	30↓		
Pea	Toasted flour (TF)	30↓		
Soy protein	Protein concentrate (SC)	2=, 3↑, 4↑	↑ DS in SC (AL), ↓ DS in 11S and 7S vs. control. ↓ DSF in SC (AL) and ↑ DSF in 11S and 7S vs. control.	[45]
7S soy protein	Soy protein fraction (7S)	2↑, 3↑, 4↑		
11S soy protein	Soy protein fraction (11S)	2↑, 3↑, 4↑		
Walnut	Flour	20↑, 30↑, 40↑, 50↑	↓ DS in all treatments vs. control. ↑ DSF at 20% and ↓ DSF in 30, 40 and 50% substitution vs. control.	[46]
Mealworm	Powder (MP)	5↓, 10↓, 15↓	↑ DDT in MP at 5 and 10%, CHP (AL) and CRP at 10%, and ↓ DDT in MP at 15% and CRP at 15% vs. control. ↑ DS and ↓ DSF in all treatments vs. control. ↑ P and ↓ L in MP (AL) and CRP (AL) and ↓ P and ↑ L in CHP at 10 vs. control. ↑ P/L in MP (AL) and CRP (AL) vs. control.	[34]
Chickpea	Powder (CHP)	5↑, 10↑, 15↑		
Cricket	Powder (CRP)	5↑, 10↑, 15↑		
Mealworm	Powder	5↓, 10↓	= DDT and = DS in MF (AL) vs. control. ↑ P/L in MF (AL) vs. control.	[20]
Cricket	Powder	10↓, 30=	Similar DDT at 10% and ↑ DDT at 30% vs. control. ↑ DS at 10% and ↓ DS at 30% vs. control. ↑ P/L in all treatments vs. control.	[19]
Strip loin beef	Powder	3, 5, 7, 10	↑ Elongation resistance, ↑ elongation and ↓ max resistance value as the additive rates increase vs. control.	[47]
Yoghurt	Crude (Yg)	30–50	↑ Dough structure, ↑ EX and ↑ deformation energy in Yg and ↓ in Cc vs. control.	[25]
Curd cheese	Crude (Cc)	30–50		
Whey protein	Concentrate (WPC)	5↓, 10↓, 15↓	↑ Arrival time and ↑ mixing tolerance in all treatments. ↓ DDT in all treatments at 10 and 15%. ↑ DS in all treatments at 5 and 10%, and ↓ DS at 15% vs. control. ↑ DSF in hydrolysates treatments and ↓ DSF in concentrates vs. control.	[31]
Whey protein	Protein hydrolyzed	5↓, 10↓, 15↓		
Casein	Protein concentrate	5↓, 10↓, 15↓		
Casein	Protein hydrolyzed	5↓, 10↓, 15↓		
Whey protein	Protein concentrate (WC)	5↓, 10↓, 15↓, 20↓, 25↓, 30↓	↓ DS in WC, ↑ DS in SC vs. control.	[42]
Soy	Protein concentrate (SC)	5↑, 10↑, 15↑, 20↑, 25↑, 30↑		
White button mushroom	Powder (WBP)	5↑, 10↑, 15↑	= DDT in WBP (AL), ↓ DDT in SMP (AL) and PMP (AL). ↓ DS in all treatments vs. control.	[41]
Shiitake mushroom	Powder (SMP)	5↑, 10↑, 15↑		
Porcini mushroom	Powder (PMP)	5↑, 10↑, 15↑		
Algae: T. chuii	Powder	4↑, 8↑, 12↑, 16↑	↓ DS and ↓ EX in all formulations vs. control.	[48]
	Extracted	4↑, 8↑, 12↑, 16↑		
Faba bean (FB) + Carob germ (CG) + Gluten (G)	Mix of Flours	FB:10 + CG: 5 + G:2.5	↓ EX, ↓ resistance to extension, ↑ total gas volume and ↑ volume gas retention vs. control.	[26]
Soy (S) + Fructooligosaccharides (FOS)	Protein hydrolysate	S: 7.73 FOS: 5.60	↑ DDT in all treatments vs. control.	[49]
		S: 17.10 FOS: 8.55		

↑, ↓ or = represent differences between treatments vs. control. Percentage of addition (%) vs. water absorption column present differences between treatments vs. control. DDT: Dough development time, DS: Doug stability time, DSF: Degree of softening, TE: Dough tenacity, EX: Dough extensibility. AL: All levels.

Dough characteristics are essential to predict the baking performance of the product. The addition of different protein sources impacts the dough, primarily by interfering with gluten bonds and increasing WA of the different blends, changing the viscoelastic parameters of the final dough.

## 4. Texture and Color in Bread Protein Enhancement

Adding different protein sources to bread affect the technological characteristics because of different protein interactions in the formulated bread. Specific volume (SV), texture profile (TP), color, or sensory characteristics may change compared to the control (Table 3), making this one of the main challenges in the improvement of protein in bread.

SV is an important bread characteristic that quantitatively measures baking performance, where light and airy bread is attractive to consumers [31]. In most cases, protein enhancement in bread will decrease loaf volume as protein addition increases because of lack of gluten-forming proteins. Additionally, protein addition can disrupt disulfide bonds formation among gluten proteins, causing less trapped air by the gluten network [21]. Conversely, SV could influence starch in vitro digestibility, where higher SV could increase the activity of amylases on starch granules [50].

Protein source, protein process, form of addition, and protein concentration will provide different results in SV. Concentration up to 20% flour or powder addition could enhance loaf volume with carob germ [12], barley, and lentil [51], in addition to ≤10% of lupine [15,44] and porcini mushroom [41]. Higher addition or different sources of proteins will decrease loaf volume.

Whey protein concentrate can be added in higher amounts, where 20–30% improves loaf volume, but concentrations lower than 15% generate lower bread volume [31,42]. Hoehnel et al. (2019) used different vegetable protein isolate sources, indicating loaf volume enhancement in gluten, zein and potato at 15% addition, although legume sources such as pea and lupine demonstrate lower loaf volumes at 15% addition [12]. Notably, some protein hydrolysates can improve loaf volume at concentrations lower than 15% additions in most cases [8,31,45,52].

The texture is essential in a bread product and manifests the structure perceived by tactile and kinesthetic senses. Texture profile analysis (TPA) is a test performed by compressing and decompressing a piece of food of a defined size and shape, placed on a base plate which is compressed twice, simulating the chewing action of the teeth. The instrument generates two curves defined by two axes: force and time. These curves represent different parameters that define the mechanical characteristics of the bread [53,54].

The mechanical characteristics of the TPA are divided into primary attributes: Hardness, cohesiveness, and springiness in bread. Secondary attributes in bread are chewiness and gumminess, derived from the calculation of primary attributes [54]. Differences in these attributes are described in Table 3, where different addition of protein sources impacts bread quality.

Hardness is a critical parameter in bread. It indicates the maximum force required to compress the bread [16]. Texture results are depicted in Table 3, where the hardness increases or decreases depending on the protein source, addition level, or flour treatment. Hydrolyzed vegetable protein sources such as soybean, maize germ, and mixes of lentil, pea, and faba bean in blends up to 30% when incorporated in bread have a similar or lower hardness compared to the control. In contrast, animal protein sources such as milk or anchovy hydrolysates create a harder bread compared to control. A similar trend is presented with flours and protein concentrates, where vegetable sources including chickpea, soy, lupin, carob germ, and some mushrooms create a softer bread than control with the addition of mixtures lower than 15%. In contrast, adding whey, casein, and salmon meal resulted in a harder bread compared to control. Vegetable sources such as pea, faba bean, amaranth, quinoa, potato, zein, hemp, and seaweed create an increased hardness compared to the control in most substitution levels. Edible insects such as grasshopper and mealworm are a good source of protein, creating a softer bread up to 20% powder substitution.

**Table 3 foods-11-02399-t003:** Bread textural and sensory characteristics.

Protein Source	Type	Percentage of Addition (%) vs. Specific Volume	Texture Characteristics	Best Conditions	Color L* Crust	Color L* Crumb	Sensory Results	Reference
Lupine	Debittered Flour (DF)	10↑, 15↓, 20↓	↑ HA in DF and FF at 15–20% vs. control. ↓ CH in all treatments except in DF at 20%. ↓ SP, ↓ CO and ↓ RE in DF and FF.	FF at 20% but acidity should be masked.	↓ L* in DF and FF vs. control. Lighter DF vs. FF.	↓ L* in DF and FF vs. control. Lighter DF vs. FF.	↓ Acceptance of FF due to their acidic taste and flavor. DF was similar vs. control.	[15]
	Fermented Flour (FF)	10↑, 15↓, 20↓						
Chickpea	Flour (CF)	15↓	Similar HA, CO and RE in all treatments vs. control.			↓ L* in SGF vs. all treatments.	Similar texture, color, odor, aroma, and overall acceptance in all treatments vs. control.	[55]
	Germinated Flour (GF)	15↓						
	Selenium-fortified germinated flour (SGF)	15↓						
Soy	Protein concentrate (SC)	5, 10, 15	↑ HA, ↑ CH and ↓CO in PC at all levels vs. control. Similar HA, CH and CO in SC vs. control. Similar SP in all treatments vs. control.	SP at 5–15%.		↓L* in al treatments vs. control. Lighter SC vs. PC.	↓ overall acceptability in all treatments. Darker crust and crumb color. Similar bitter and astringent flavors,↑ HA, ↑ adherence, ↑ GUM and ↑ CH in all treatments vs. control.	[11]
Pea	Protein concentrate (PC)	5, 10, 15						
Pea	Flour	30↓	↑ HA in all treatments vs. control. ↓ SP, ↓ CO and ↓ RE in all treatments vs. control.	TF at 30%.	↓ L* vs. control, were GRF was the darker.	↑ L* vs. control.		[14]
	Germinated Flour (GRF)	30↓						
	Toasted flour (TF)	30↓						
Faba bean	Sourdough/Flour (SRD)	30 ↓	↑ HA, ↓ CO, ↓ SP, ↓ CH and ↓ RE in FBF and SRD vs. control, where SRD was the hardest.	FBF at 30%.				[56]
	Flour (FBF)	30 ↓						
Lupine	Flour (LF)	3↓, 6↓	↑ HA, ↑ CH in LF at 3 and 6% vs. control. ↓ HA in FLF at 3 and 6% and ↓ CH in FLF at 3%. Similar SP and RE in LF, and ↓ SP and ↓ RE in FLF vs. control.	FLF at 3%.			↑ color, flavor, and acidity in all treatments vs. control. ↓ taste in LF vs. control, but ↑ taste in FLF.	[44]
	Fermented Flour (FLF)	3↑, 6↑						
Soy	Protein hydrolysate (SH)	20↓	Similar HA vs. control.	SH at 20%.		↓ L* vs. control		[49]
Soy protein	Protein concentrate (CP)	2↓, 3↓, 4↓	↑ HA and ↑ CH in CP and 7S at 4% vs. control. Similar CO in AL except in 11S and 7S at 4% were it was lower.	<4% of 11S soy protein fraction.			↑ Exterior appearance and structure in 11S 2% and 3% vs. control, and similar at 4%. All treatments had similar taste and flavor. 11S at 3% was the best scored.	[45]
	Protein fraction (7S)	2↑, 3↑, 4↑						
	Protein fraction (11S)	2↑, 3↑, 4↑						
Amaranth	Flour (AF)	5↓, 10↓, 15↓	↑ HA and ↑ CH at 5, 10 and 15% vs. control. =SP and = CO in AL vs. control.	AF at 10%.	= L* vs. control.	= L* vs. control.	The best in overall acceptability was the control, then 5%, and the lower acceptability was 15% substitution.	[16]
Amaranth: *A. spinosus*	Flour (ASF)	25↓, 50↓	↑ HA, ↑ GUM and ↑ CH in all treatments vs. control. Similar CO in all treatments vs. control. ↑ SP in ASF and AHF at 25% vs. control.	AHF at 25%.	↓ L* vs. control	↓ L* vs. control	All treatments indicate lower scores vs. control, were AHF present better acceptability vs. ASF.	[17]
Amaranth: *A. hypochondriacus*	Flour (AHF)	25↓, 50↓						
Quinoa	Flour (QF)	5↓, 10↓, 15↓	↑ HA at all levels vs. control. ↓ SP, ↓ CO and ↓ RE at all levels vs. control.	QF at 10%.				[9]
Maize germ protein	Protein hydrolysate (MGPH)	1↑, 2↑, 4↑	Similar HA, CO and CO at 1% vs. control. ↓ HA, ↑ CO, ↓CH at 2 and 4% vs. control. Similar SP at all levels.	MGPH at 4%.	↓ L* vs. control.	= L* vs. control.	Similar color, taste, chewability, texture vs. control. ↓ Aroma score vs. control.	[8]
Gluten	Protein isolate (GI)	15↑	↓ HA in GI and CF vs. control. ↑ HA in the other treatments vs. control.	PI and GI at 15%.	↓ L* in all treatments, except ZI vs. control.			[12]
Zein	Protein isolate (ZI)	15↑						
Potato	Protein isolate (PI)	15↑						
Carob germ	Flour (CF)	15↑						
Pea	Protein isolate (PI)	15↓						
Lupine	Protein isolate (LI)	15↓						
Faba bean	Flour (FBF)	15↓						
Barley	Sourdough/Raw Flour (BRS)	20↑	↑ HA in all treatments, were quinoa present the hardest treatment. ↓ RE in all treatments vs. controls were chickpea present the lower value.	Barley and lentil treatments.			↑ Global index of the palatability, was higher in controls, BRS, BSS, and LSS, and lower in CRS, CSS, and QSS. In particular, the most appreciated bread was the control sourdough, while the lowest score corresponded to CSS.	[51]
	Sourdough/Sprouted Flour (BSS)	20↓						
Chickpea	Sourdough/Raw Flour (CRS)	20↓						
	Sourdough/Sprouted Flour (CSS)	20↓						
Lentil	Sourdough/Raw Flour (LRS)	20↑						
	Sourdough/Sprouted Flour (LSS)	20↓						
Quinoa	Sourdough/Raw Flour (QRS)	20↓						
	Sourdough/Sprouted Flour (QSS)	20↓						
Walnut	Flour (WF)	20↓, 30↓, 40↓, 50↓		WF at 30%.			↑ Overall acceptability 10% and 20% vs. control. ↓ crumb color score as addition increased. ↑ Crumb texture, taste and flavor in 10% and 20%, ↓ in 30% and 40%.	[46]
Apricot kernel	Flour (APF)	4↓, 8↓, 12↓, 24↓	↑ HA as addition increased vs. control, except at 5% substitution. Similar SP and CO in 4, 8, and 12%, and ↓ SP and CO at 24% vs. control.	APF at 8%.	↓ L* vs. control.	↓ L* vs. control.	Similar appearance, smell, crust color, taste, texture, and overall acceptability of bread at 4 an 8% vs. control.	[10]
Hemp	Sourdough/Flour (HSS)	5↑, 10↓, 15↓	↑ HA all treatments vs. sourdough control. Similar RE in all treatments vs. sourdough control.	HSS at 10%.			Good sensory and texture properties still remain. Overall taste increased according to the amount HSS used.	[18]
Chia seed	Flour (CHF)	2, 4, 6	↓ HA in CHC at 4 and 6% and CHF at 4% vs. control. ↑ HA in CHF at 2 and 4%, and CHC at 2% vs. control.	CHF at 6%.		↓ L* vs. control. ↑ L* CHC vs. CHF.	All samples present better values vs. control. 2% chia powder was the best.	[57]
	Cakes (CHC)	2, 4, 6						
Grasshopper	Powder (GP)	10↓, 20↓	↓ HA and ↓ SP in all treatments vs. control. = CO in GP (AL) and GDP (AL) vs. control.	GP at 10%.			↓ Overall preference in GP at 20% and GDP at 20% vs. control. Similar in GP at 10% to control.	[21]
Grasshopper	Defatted Powder (GDP)	20↓						
Mealworm	Powder (MP)	5, 10	↓ HA in all treatments vs. control.	MP at 5%.	↓ L* vs. control	↓ L* vs. control	↓ Overall linking in all treatments vs. control.	[20]
Mealworm	Sourdough/Powder (MS)	5, 10						
Cricket	Powder (CP)	10, 30		CP at 10%.			↓ Global linking score in all treatments vs. control. CS and CP at 30% present the lowest scores.	[19]
	Sourdough/Powder (CS)	10, 30						
Cinereous cockroach	Powder (CIP)	5↓, 10↓, 15↓	↑ HA as addition increased vs. control.	CIP at 10%.	↓ L* vs. control.	↓ L* vs. control.	↓ Total score from external and internal characteristics, aroma, and taste in CIP at all levels vs. control.	[22]
Yoghurt	Crude (YG)	10↓, 20↓, 30↓, 50↓, 70↓	↓ HA in YG 10 -50% vs. control, but ↑ HA at 70 vs. control. ↑ HA in CC at all levels.	YG at 50%.			↑ Overall acceptability, color, flavor, taste, texture, and appearance in YG 50% and CC 30% addition vs. control.	[25]
Curd cheese	Crude (CC)	10↓, 20↓, 30↓, 50↓, 70↓						
Whey protein	Protein concentrate (WC)	5↓, 10↓, 15↓, 20↑, 25↑, 30↑	↑ HA and ↑ CH in WC(AL) and SC at 15–30%, but ↓ HA and ↓ CH in SC at 5 and 10% vs. control. ↑ CO and ↑ GUM in WP at 15–30%. ↓ CO and ↑ GUM at SC at 25 and 30%. ↑ RE in WC at 15–30% and SC at 5–20%. And ↓ RE in WC at 5–10% and SC at SC at 25–30%. Similar SP in all treatments, except in SC at 20–30%.	WC and SC at 15%.	↓ L* vs. control	↓ L* vs. control		[42]
Soy	Protein concentrate (SC)	5↓, 10↓, 15↓, 20↓, 25↓, 30↓						
Whey protein	Protein concentrate (WPC)	5↓, 10↓, 15↓	↑ HA in all treatments vs. control.	5% level incorporation of both milk treatments.	Darker vs. control.	Darker vs. control.	↓ Overall acceptability, crust and crumb color, texture, and flavor in all treatments vs. control.	[31]
	Protein hydrolyzed (WPH)	5↑, 10↑, 15↑						
Casein	Protein concentrate (CAC)	5↓, 10↓, 15↓						
	Protein hydrolyzed (CAH)	5↑, 10↑, 15↑						
Strip loin beef	Powder (SLBP)	3, 5, 7, 10		SLBP at 3%.		↓ L* vs. control.	↓ Overall acceptability at all levels vs. control.	[47]
Labeobarbus fish	Powder (LP)	5, 10, 15, 20		LP at 10%			Similar overall acceptability at 5 and 10% vs. control. Similar color, texture, and taste in 5 and 10% vs. control. ↑ flavor score in 5 and 10% vs. control.	[58]
Anchovy	Protein hydrolyzed (AH)	1.46↑, 2.93↑, 5.85↑, 11.7↓	↑ HA and ↓ adhesiveness as substitution increased vs. control.	AH at 1.46%.			Higher AH concentrations indicated ↑ saltiness and sourness, but lower sweetness, crust color, crumb color, and moisture.	[52]
Salmon: *Oncorhynchus tschawytscha*	Powder (SFP)	5↓, 10↓, 15↓	↑ HA in all levels vs. control. ↓ CO, ↓ RE and ↓ SP all llevels vs. control. ↑ GUM and ↑ CH in 5 and 10% substitution, but ↓ GUM and ↓ CH 15% substitution vs. control.	SFP at 15%.	↓ L* vs. control.	↑ L* vs. control.		[23]
Tilapia-waste	Powder (TP)	2.5, 5, 10, 15, 20		TP at 5–10%.			↓ Overall linking in all levels vs. control. TP at >20% caused changes in sensory characteristics including appearance, aroma, flavor/taste, texture, and mouthfeel.	[24]
White button mushroom	Powder (WBP)	5↓, 10↓, 15↓	↑ HA in WBP(AL) and SMP at 10 and 15% vs. control. ↓ HA in PMP(AL) and SMF at 5%. ↓ SP in all treatments vs. control. ↓ GUM in WBP at 5 and 10%, SMP at 5% and PMP(AL) vs. control.	PMP at 10%.				[41]
Shiitake mushroom	Powder (SMP)	5↓, 10↓, 15↓						
Porcini mushroom	Powder (PMP)	5↑, 10↑, 15↓						
Algae: T. chuii	Powder (AP)	4↓, 8↓, 12↓, 16	↑ HA in all treatments vs. control. ↑ HA in AP vs. AE.	AP and AE at 12%				[48]
	Extracted (AE)	4↓, 8↓, 12↓, 16↓						
Defatted soy (DSF) + Whey protein (WPC)	Mixed flour and powder	DSF:8.2 + WPC:3	↑ HA, ↑ CH and ↓ CO vs. control. Similar SP vs. control.	Mix of 88.8% wheat flour, 8.2% of DSF and 3% of WPC.			90% of participants had positive responses.	[28]
Lentin (L) + Pea (P) + Faba bean (FB)	Protein hydrolysate	L: 10 + P: 10 + FB: 10↓	Similar HA, RE and fracturability in 30% addition vs. controls.	Mix legume hydrolysate addition at 30%.	↓ L* vs. controls.		Sensory analysis demonstrated that the legume flours hydrolysate did not modify the scores vs. control.	[59]

↑, ↓ or = represent differences between treatments vs. control. Percentage of addition (%) vs. Specific Volume column present differences between treatments vs. control. TPA: Texture Profile Analysis. HA: Hardness and firmness, CH: Chewiness, CO: Cohesiveness, SP: Springiness, RE: Resilience, GUM: Gumminess, L*: Lightness. AL: All levels.

The different textures related to hardness are possibly due to water competition between protein and starch. A lower starch gelatinization may generate increased hardness, [8,12] but starch digestibility decreases due to the relationship between swelling and hydration starch with the chemical activity between starch and amylases [60]. Soy and whey proteins have created harder products compared to their respective control, where whey protein addition creates higher hardness than soy protein, suggesting that whey protein has higher gelling properties compared to soy protein [42].

Hardness increases considerably through time due to starch retrogradation. The addition of hydrolysates in bread might reduce staling process. Competition for available water between hydrophilic groups of hydrolysates with amylose or amylopectin during gelatinization could reduce starch gelatinization and lower crystal formation during retrogradation [8]. Previous acidification of lupin flour by lactic fermentation changed the functional properties by increasing water absorption when the dough was integrated, reinforcing the gluten network and increasing the extension of resistance during fermentation, generating a softer bread [44]. The integration of insects in bread can change the texture parameters, such as hardness and elasticity, due to the amount of insect fat that plasticizes and lubricates the dough, increasing the incorporation of air during kneading [20,33].

Bread cohesiveness refers to how the dough holds together during chewing [54]. The addition of different protein sources lightly affects this parameter. As protein addition increases, cohesiveness in bread decreases.

Springiness as resilience indicates the ability of bread to recover after deformation due to compression, where springiness is represented as millimeters and resilience as a ratio dividing the upstroke energy of the first compression by the downstroke energy of the second compression [23,54]. Like cohesiveness, with the addition of protein, springiness and resilience decrease. Furthermore, it is also related to volume, where the more gas trapped in the bread alveolus, the faster it will recover quickly after compression [21]. Millar et al. (2019) suggest that the reduction of springiness, cohesiveness and the increase in crumb hardness could be produced by the reduction in moisture content of the crumb [14].

Chewiness represents the energy needed to disintegrate bread structure ready to swallow, like gumminess, where chewiness is used for solid foods and gumminess for semisolid foods. Some foods, such as cake or bread, become semisolid during mastication; this is the reason for some authors to use both [53]. In the TPA method, gumminess is related to hardness and areas during compression, and chewiness is related to gumminess value and the duration of compression. In Table 3, the additions of different protein sources indicate how hardness is related to chewiness and gumminess in most treatments. However, in some treatments, the addition of lupine, faba bean, corn protein hydrolysate, salmon, and mushroom did not demonstrate the same trend.

The appearance in terms of color formation in the “golden yellow” or brown crust of bread is called “browning”. It results from two different chemical reactions: Maillard reaction, and caramelization during the baking process. The final color may vary depending on the bread product, raw materials and their processing, protein source, and the baking process conditions (cooking time and temperature) [20,61]. In Table 3, adding different protein sources will generate darker (lower Luminosity values) crust color and darker crumb color. This darker color is related to the interaction of amino groups with reducing sugars in the Maillard reactions while melanoidins are formed [8]. In the sourdough process it is normal to find darker results due to the proteolytic activity of lactic acid bacteria (LAB) releasing more free amino acids [20].

Evaluating a final product through the senses is important to perceive consumer acceptance. Table 3 indicates how adding different protein sources under different concentrations could generate several sensory perceptions and overall acceptance. Sourdough breads were characterized by a sour taste and odor, usually caused by LAB [18,51]. Adding vegetable ingredients may generate unpleasant characteristics, such as a grassy, musty odor, or a bitter taste [18,51,57].

Bread made with animal protein sources indicates lower acceptability, except when yogurt was added [24,25,31,52]. The increase in protein content promotes the formation of Maillard reactions during heating, forming pyrazines and pyrolines, causing different notes in flavors and aroma. In addition, compounds responsible for bread odor, such as acetaldehyde, are produced during fermentation [8]. In most cases, adding protein sources to bread did not impact consumer preference for taste, but wheat bread is still better accepted.

Bread nutrition enhancement is essential to new trends of healthy products, but characteristics in bread, such as texture, color, and aroma, are important parameters to consumer’s acceptance. Generating new methods of protein addition or processes that can provide attractive features and good taste in bread products should be explored in future research.

## 5. Health Benefits of Enriched Bread

Starch digestibility and glycemic response of bread in the human body are affected by the matrix of food components such as proteins, lipids, and carbohydrates [23], and food processing, such as the use of sourdough or germinated flour altering the rate of starch hydrolysis related to glycemic index [18,51,56]. Different additions of protein ingredients could reduce starch digestibility and decrease the glycemic index (Table 4).

The reduction of starch digestion is related to low glycemic index foods. Peptides in the protein hydrolysate have an inhibitory effect on α-amylase. These hydrolysates could interrupt the interaction of the enzyme with the substrate binding to the enzyme active site. The hydrolysates could resist gastrointestinal conditions by the same mechanism [8]. Also, polyphenols could inhibit digestive enzymes [9]. To do that, the addition of protein hydrolysates in bread could have a positive effect on reducing sugar release.

A reduction of starch digestion by salmon powder addition may be observed. This could be attributed to the increased protein and lipid content that can form an amylose–lipid complex. Those interactions can resist starch digestion [23]. Additionally, reduced enzymatic activity to hydrolyze starch granules has been reported due to presence of proteins that could encapsulate starch granules as a protective barrier against amylose hydrolysis [23,50].The use of sourdough minimizes the rate of starch hydrolysis due to the acidification generated [18]. In contrast, using germinated flour may increase the sugar content, which increases the estimated glycemic index compared to using raw ingredients [51].

Table 4 depicts how protein digestibility changes as a function of pretreatment and protein source. A reduction of protein digestibility could be due to the formation of protein–starch complex, cross-links between proteins, the presence of fiber and minerals in the bread matrix, and the interaction of protein with antinutritional factors, such as tannin, phytates, saponins, and phenolic compounds [62,63,64]. Sourdough can increase protein digestibility in response to proteolysis and inactivation of some antinutritional factors, such as trypsin inhibitor and condensed tannins by LAB [18,56]. The combination of germinated flour and sourdough can reduce protein complexes and antinutritional factors, such as improving proteolysis by increasing the free amino acid content, including gamma-aminobutyric acid (GABA) [51]. Something different happens with only grain germination is used; protein digestibility is slightly reduced compared to non-germinated [55].

Some antinutritional factors, such as phytates, also have beneficial effects, including antioxidant activity, anticarcinogenic potential and prevention of heart diseases [17]. Total phenols are highly correlated with antioxidant activity in food products [9,51]. Wheat bread blended with grains and seeds such as quinoa, barley, lentil, chia, chickpea, salmon or maize hydrolysates indicates antioxidant potential (Table 4).

The grain germination process could improve antioxidant activity due to the higher liberation of phenols [9,51]. The addition of maize protein hydrolysates increases the antioxidant activity of formulated bread. However, the thermal process affects the stability and bioactivity of those molecules [8].

Currently, consumers are looking for healthy products. Integrating different protein sources or processes is fundamental to increasing the health potential of bread products and achieving the functional effect needed. Keeping innovating in the process of protein ingredients and bread making is fundamental to satisfying the market demands.

## 6. Conclusions

The selection of protein sources, the amount of addition, and the dough processing and baking methods are factors to consider when producing protein-rich bread. The addition of different protein sources influences from competition for water with gluten to form the dough network structure, due to competition for water with gluten to form, which affects carbon dioxide retention during fermentation. It can also impact crumb characteristics and sponginess during baking, which are associated with the texture and appearance of the bread.

Depending on the protein source, it could interact with the other ingredients, impacting the color of the bread crust and the taste of bread, influencing consumer acceptance. Therefore, processing strategies such as sourdough formation may favor the development of bread with improved textural properties, flavor, and protein digestibility benefits.

Conversely, incorporating protein sources with functional properties in bread could provide health benefits, such as antioxidant potential, lower glycemic index, and enrichment in micronutrients such as vitamins and minerals. The lowering of glycemic index is related to the resistance of starch hydrolysis by digestive enzymes in response to different interactions between lipids, proteins, and starch, and the effect of polyphenols among digestive enzymes. The protein digestibility parameter changes among treatments, where sourdough could be an excellent alternative to inactivate antinutritional factors.

Having technofunctional, sensory, and functional characteristics in bread product is a great challenge for the food industry. Therefore, it is necessary to search for new strategies to incorporate different protein sources not only to improve the nutritional profile of bread but also to have a product with high quality characteristics, including good texture profile and taste that could provide health benefits to consumers.

## Figures and Tables

**Figure 1 foods-11-02399-f001:**
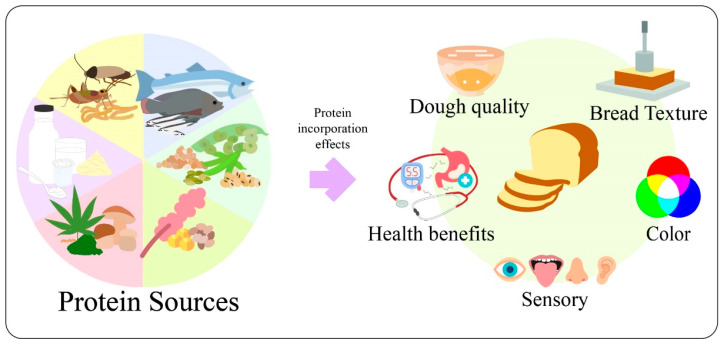
Protein sources (milk derivates, edible insects, fish derivates, legumes, cereals, pseudocereals, and other sources) and their effects on mass, process, final product, and health.

**Table 1 foods-11-02399-t001:** Nutritional content in 100 g of bread (dry weight).

Protein Source	Type	Percentage of Addition (%)	Energy (kcal)	Macronutrients (g)			Fiber	Ash	Reference
				Protein	Lipids	Carbohydrates			
Legumes									
Pea	Flour	30	399.1	15.6	2.8	78.0	ND	3.7	[14]
	Germinated Flour	30	399.9	16.1	2.8	77.5	ND	3.6	
	Toasted flour	30	399.7	16.0	2.8	77.8	ND	3.5	
Pea	Protein concentrate	5	400.5	17.3	1.5	79.5	ND	1.7	[11]
		10	399.6	21.7	1.5	74.8	ND	2.0	
		15	398.8	25.4	1.6	70.7	ND	2.3	
Soy	Protein concentrate	5	401.7	15.4	2.0	80.5	ND	2.1	
		10	404.6	17.0	2.6	78.3	ND	2.1	
		15	404.9	18.6	2.9	76.0	ND	2.4	
Lupin	Debittered Flour	10	418.1	18.5	7.8	71.2	2.7	2.5	[15]
		15	416.9	20.0	8.0	69.4	3.2	2.6	
		20	424.6	20.4	10.0	67.0	3.6	2.7	
	Fermented Flour	10	418.4	17.6	7.5	72.4	2.3	2.5	
		15	418.7	19.9	8.1	69.3	2.9	2.6	
		20	424.3	20.1	9.7	67.6	3.4	2.7	
Pseudocereals									
Amaranth	Flour	5	402.9	18.5	3.4	76.4	1.9	1.7	[16]
		12	403.4	19.2	3.9	75.0	2.0	2.0	
		15	401.4	19.6	4.1	74.0	2.6	2.3	
Amaranth: *A. spinosus*	Flour	25	353.5	18.1	0.7	78.2	9.5	3.0	[17]
		50	346.5	18.4	1.5	76.7	11.7	3.5	
Amaranth: *A. hypochondriacus*	Flour	25	352.3	17.6	0.8	78.6	9.9	3.0	
		50	346.5	19.0	1.5	75.5	11.3	4.0	
Other seeds									
Apricot kernel	Flour	4	428.0	17.6	7.3	73.1	ND	2.1	[10]
		8	451.0	17.7	12.0	67.9	ND	2.3	
		15	497.6	20.2	21.5	55.9	ND	2.4	
		24	507.4	21.2	23.8	52.2	ND	2.9	
Hemp	Sourdough/Flour	5	376.5	13.0	1.3	85.4	7.2	0.8	[18]
		10	368.5	14.1	1.7	83.0	8.8	1.1	
		15	366.5	15.2	2.1	81.4	9.7	1.3	
Insects									
Cricket	Powder	10	388.5	37.7	2.1	58.0	3.4	2.1	[19]
		30	425.2	45.7	10.5	41.2	4.3	2.6	
	Sourdough/Powder	10	416.6	35.6	7.1	55.4	2.8	1.9	
		30	419.7	42.0	8.4	47.2	3.3	2.3	
Mealworm	Defatted powder	5	400.4	14.9	0.7	83.7	ND	0.7	[20]
		10	404.3	16.7	1.6	80.7	ND	0.9	
	Sourdough/Defatted powder	5	400.4	14.9	0.7	83.7	ND	0.7	
		10	403.0	16.6	1.3	81.2	ND	0.9	
Grasshopper	Powder	10	394.2	14.8	1.8	81.0	1.3	2.4	[21]
		20	398.2	17.4	3.0	77.0	1.6	2.6	
	Defatted powder	20	388.8	18.1	1.2	78.1	1.7	2.6	
Cinereous cockroach	Flour	10	399.9	22.7	5.6	67.1	2.3	4.7	[22]
Fish									
Salmon: *Oncorhynchus tschawytscha*	Powder	5	420.8	16.3	6.0	75.4	ND	2.4	[23]
		10	426.6	18.2	7.3	72.1	ND	2.4	
		15	436.0	20.0	9.1	68.4	ND	2.4	
Tilapia-waste	Powder	2.5	369.9	12.3	2.6	82.5	8.2	2.6	[24]
		5	372.0	15.6	3.6	77.7	8.4	3.1	
		10	372.4	17.5	3.9	75.3	8.4	3.3	
		15	371.6	22.7	4.7	68.8	9.3	3.7	
		20	372.5	25.6	5.5	64.3	9.2	4.6	
Milk products									
Yoghurt	Raw	30	405.9	14.9	3.9	77.7	ND	3.4	[25]
		50	405.6	15.8	4.3	75.9	ND	4.0	
Curd cheese	Raw	30	425.8	17.1	8.8	69.6	ND	4.5	
		50	452.5	20.3	14.8	59.6	ND	5.3	
Mixes									
Faba bean (FB) + Carob germ (CG) + Gluten (G)	Flour	FB:10 + CG: 5 + G:2.5	377.3	22.8	2.2	71.5	4.9	3.5	[26]
Cassava (CF) + Soy bean (SF)	Flour	CF:10 + SF:19	385.6	17.5	8.0	64.6	3.6	8.0	[27]
Soy (SDF) + Whey protein (WPC)	Defatted flour; Protein concentrate	SDF:8.2 + WPC:3	381.2	13.9	3.0	80.1	5.5	2.9	[28]

Carbohydrates: Total carbohydrates. Proteins, lipids, carbohydrates, fiber, and ash expressed on dry basis. ND: Not determined.

**Table 4 foods-11-02399-t004:** Potential health benefits.

Protein Source	Type	Percentage of Addition (%)	Protein Digestibility	Health Benefits	References
Faba bean	Flour	30	63.6↓	↓ Predicted glycemic index (eGI) in sourdough treatment (84.2) vs. flour treatment (91.4) and control (94.6)	[56]
	Sourdough/Flour	30	74.1↑		
Amaranth: *A. spinosus*	Flour	25, 50		↑ Mineral content increased and ↑ increase of phytic acid content with the inclusion of amaranth flour in the bread.	[17]
Amaranth: *A. hypochondriacus*	Flour	25, 50			
Quinoa	Flour	5, 10, 15		↑ TPC, ↑ radical scavenging capacity (ABTS, DPPH and HOSC), ↓ HI, ↓ eGI, ↓RDS and ↑SDS by flour addition.	[9]
Maize germ protein	Protein hydrolysate	1, 2, 4		↑ DPPH radical scavenging, ↑ Fe2+ chelating activity and ↓ starch digestion at 20 min of digestion (in vitro digestion) related to the effect of peptides in the hydrolysate.	[8]
Barley	Sourdough/Raw Flour	20	65.5↓ *	↑ TPC and radical scavenging activity in barley, lentin and quinoa vs. control. ↑ TPC and radical scavenging in sourdoughs treatments. ↓ Condensed tannins, trypsin inhibitor activity, and α-galactosides, in sourdoughs. ↓ Phytic acid after fermentation, except in barley flour. ↓ eGI in all breads in all treatments, except in quinoa samples, were barley and chickpea treatment demonstrate the lowest GI. ↑ GABA in cereals, pseudocereals, and legumes.	[51]
	Sourdough/Sprouted Flour	20	72.8↑ **		
Chickpea	Sourdough/Raw Flour	20	74.8↑ *		
	Sourdough/Sprouted Flour	20	76.8↑ **		
Lentin	Sourdough/Raw Flour	20	73.3↑ *		
	Sourdough/Sprouted Flour	20	74.0↑ **		
Quinoa	Sourdough/Raw Flour	20	64.6↓ *		
	Sourdough/Sprouted Flour	20	75.1↑ **		
Hemp	Sourdough/Flour	5 10 15	87.6 = 88.4 = 88.1 =	↓ Hydrolysis index, ↓ Predicted glycemic index, ↓ phytic acid and ↓ total saponins in hemp sourdough treatment.	[18]
Chia seed	Flour	2, 4, 6		↑ TPC and ↑ TEAC in fortified bread by chia seed.	[57]
	Cakes				
Salmon: *Oncorhynchus tschawytscha*	Powder	5 10 15	80.8↑ 80.2↑ 80.6↑	↓ Starch digestion. ↓ TPC and ↑ total antioxidant activities by protein-phenolic of phenolic-lipid complexes.	[23]
Faba bean (FB) + Carob germ (CG) + Gluten (G)	Mix of Flours	FB:10 + CG: 5 + G:2.5	88.2↓	↑ Antioxidant potential by phenolic acids and flavonoids in wholegrain flours.	[26]
Cassava (CF) + Soy bean (SF)	Flour	CF:10 + SF:19		↑ Phytate and ↑ condensed tannin content by flour addition.	[27]

↑, ↓ or = represent differences between treatments vs. control. Protein digestibility column present differences between treatments vs. control product. TPC: Total phenolic content, DPPH: Free radical method 2,2-diphenyl-1-picryl-hydrazil-hydrate, FRAP: Ferric antioxidant power. TEAC: Total antioxidant activities, ABTS: Scavenging capacity, HOSC:·OH Scavenging capacity, HI: Hydrolysis index, eGI: Predicted glycemic index, RDS: Rapidly digestible starch and SDS: Slowly digestible starch. IVPD: In vitro protein digestibility. GABA: A non-protein amino acid. * Compared with raw wheat flour sourdough. ** Compared with sprouted wheat flour sourdough.

## Data Availability

Data will be available upon request.

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
