# Peer review of "Protein Ingredients in Bread: Technological, Textural and Health Implications"

_foods, 2022, doi:10.3390/foods11162399_

Round 1

Reviewer 1 Report

The manuscript is well structured on the scientific level as well as on cultural relevance. It may represent a valid reference point for implementing protocols for enriched protein bakery products.

The manuscript is well structured on the scientific level and on the cultural dissemination level too. It may represent a valid reference point for the implementation of protocols for enriched protein bakery products.

The authors could consider strategic for a better qualify of the manuscript, the implementation of section 5, to argue above the glycemic index aspects. In the scientific literature, some papers describe how gluten proteins and their rheological proprieties, may affect the glycemic response. Moreover, it may be helpful to argue about the low lysine content of wheat protein and how enriching flour with other protein sources, could improve the nutritional quality.

Author Response

Reviewer 1

The manuscript is well structured on the scientific level as well as on cultural relevance. It may represent a valid reference point for implementing protocols for enriched protein bakery products.

The manuscript is well structured on the scientific level and on the cultural dissemination level too. It may represent a valid reference point for the implementation of protocols for enriched protein bakery products.

The authors could consider strategic for a better qualify of the manuscript, the implementation of section 5, to argue above the glycemic index aspects. In the scientific literature, some papers describe how gluten proteins and their rheological proprieties, may affect the glycemic response. Moreover, it may be helpful to argue about the low lysine content of wheat protein and how enriching flour with other protein sources, could improve the nutritional quality.

Author response

Thank you very much for your comments.

Regarding the glycemic index and starch hydrolysis information was added:

Lines 238 - 239:

On the other hand, SV could influence starch in vitro digestibility, where higher SV could increase the activity of amylases on starch granules [50].

Lines 279 - 281:

A lower starch gelatinization may generate increased hardness, but starch digestibility decreases due to the relationship between swelling and hydration starch with the chemical activity between starch and amylases.

Lines 345 - 348:

Starch digestibility and glycemic response of bread in the human body are affected by the matrix of food components such as proteins, lipids and carbohydrates [34], and food processing such as the use of sourdough or germinated flour altering the rate of starch hydrolysis related to glycemic index [29,51,53].

Lines 361 - 364:

Additionally, reduced enzymatic activity to hydrolyze starch granules has been reported due to the presence of proteins that could encapsulate starch granules as a protective barrier against amylose hydrolysis [34,50].

References:

Lau, E.; Soong, Y.Y.; Zhou, W.; Henry, J. Can Bread Processing Conditions Alter Glycaemic Response? Food Chem. 2015, 173, 250–256, doi:10.1016/j.foodchem.2014.10.040.

Liu, X.; Mu, T.; Sun, H.; Zhang, M.; Chen, J.; Fauconnier, M.L. Comparative Study of the Nutritional Quality of Potato–Wheat Steamed and Baked Breads Made with Four Potato Flour Cultivars. Int. J. Food Sci. Nutr. 2017, 68, 167–178, doi:10.1080/09637486.2016.1226272.

Regarding low lysine content of wheat protein, the following information was incorporated:

The information was modified:

Lines 41 – 47:

Enhancing nutrition in bread is an interesting opportunity for the food industry. Most of the bread are made with refined flour because it generates a high loaf volume, light color, homogeneous crumb porosity and soft crumb. This bread are highly accepted by costumers, even they lacks of vitamins, minerals, lysine, dietary fiber, antioxidant components and a high glycemic index [2,8,9]. This deficiency can be improved with whole wheat flour, but consumers prefer refined wheat bread [2] or other alternatives for better nutritional balance. 

Other sources are mentioned:

Line 89 – 97:

Amino acid profile in legumes contains a low level of sulfur amino acids but a high level of lysine, which could be an excellent complement to the amino acid profile of wheat flour, which lacks lysine [14]. Similarly, pseudocereals amino acid profile complements lack-lysine wheat flour. Moreover, pseudocereal and legume ingredients are rich in starch, fiber, micronutrients, and phytochemicals with potential health benefits [15]. On the other hand, animal sources such as whey and casein proteins are used in bread. Likewise, they have good technofunctional properties, including water holding capacity, solubility, and gelling properties. Also, present an excellent amino acid profile rich in lysine, methionine and tryptophan [16].

Reviewer 2 Report

The article addresses the important issue of the effect of protein additives on bread/bakery products on the quality and health-promoting properties of the resulting product.

It can be seen that the article was originally directed mainly at discussing protein additives to a typical wheat bread made from refined soft wheat flour. And this is how the narrative should be conducted. There is no need to extend the scope of the article to bread if all the examples cited and the vast majority of the literature are about bread.

It would also be appropriate to change the abstract; wheat is not the only raw material for bread; it is the worst for health, especially for bread made from highly purified flour.

And in this direction, I suggest changing the abstract and the introduction because at the moment it is not reliable data only a one-sided perspective of the Authors.

Please also change the use of the term flour for powdered ingredients like vegetable and fruit waste, edible insects or animal waste - not all powder is flour. These ingredients are powder, not flour.

The reason why bread made from so-called white wheat flour is not nutritious is not because of the wheat but because of the use of highly refined flour. A simple solution to this problem is to use whole-wheat flour - the technological quality will deteriorate, but the nutritional quality will increase significantly - the same happens when protein additives are used.

So the authors should change the main context of the narrative and justify the advantage of using protein additives over making bread from whole grain flour, currently, the narrative justifying the use of additives is laconic and naive.

Specific comments:

line 48 - easy to obtain using wholegrain flour;

line 64-66 - the bakery or only bread industry?, this line is a good example of non-justified generalisation; moreover in fact bakery industry is focused on delivering a lot of innovative and healthier products - so this line lacks sense because is too generalising

line 74 - healthier alternatives? Bread made from wholemeal flour is super healthy; you need to change the narrative in the whole article.

line 86 - this sentence is obvious; rewrite.

line 103-104 - cite a reference for edible insects versus meat-nutritional comparison.

line 106 - 108 - check the grammar of the sentence

Table 1 - for Lupin - should be debittered, not debittering

Table 1 and all tables - remove flour when not obtained from starchy grains - use the term powder instead

line 312-313 - the darker colour of the bread is not only the result of Maillard but also the innate colour of ingredients contributes to the final score.

Author Response

Reviewer 2

Extensive editing of English language and style required

Author response:

The English style of the manuscript has been revised and improved.

The article addresses the important issue of the effect of protein additives on bread/bakery products on the quality and health-promoting properties of the resulting product.

It can be seen that the article was originally directed mainly at discussing protein additives to a typical wheat bread made from refined soft wheat flour. And this is how the narrative should be conducted. There is no need to extend the scope of the article to bread if all the examples cited and the vast majority of the literature are about bread.

Author response:

Thank you very much for your comments.

The word “bakery” was changed to” bread”, and “refined wheat bread”

Line 2: In title: Review Protein ingredients in bread: technological, textural and health implications

Line 61: bread industry

Line 86: bread products

Line 95: in bread

Line 253: bread product

Line 343: bread products

Line 389: bread products

It was also considered that the narrative should focus on the term "refined wheat bread".

Line 63 – 67:

The objective of this review is to provide current information on the enrichment of refined wheat bread with protein ingredients from different sources, how it affects the dough behavior and technological properties of the bread matrix, as well as the impact of novel ingredients on sensory characteristics and the biologic potential of nutritionally improved bread.

It would also be appropriate to change the abstract; wheat is not the only raw material for bread; it is the worst for health, especially for bread made from highly purified flour.

And in this direction, I suggest changing the abstract and the introduction because at the moment it is not reliable data only a one-sided perspective of the Authors.

Author response:

The abstract was modified:

The current lifestyle and trend for healthier foods has generated a growing consumer interest in acquiring bread products with a better nutritional composition.  Mainly products with high protein and fiber and low fat. Incorporating different protein sources as functional ingredients have improved the nutritional profile but can also affect the dough properties and final characteristics of bread. This review focuses on the incorporation of different animal, vegetable, and mixed protein sources and the percentage of protein addition, analyzing nutritional changes and their impact on dough properties and different texture parameters, appearance and their impact on bread flavor and health-related effects. Alternative processing technologies such as germination and sourdough-based technologies are discussed. Using fermented doughs can improve the nutritional composition and properties of the dough, impacting positively the texture, appearance, flavor, and aroma of bread. It is essential to innovate in alternative protein sources in combination with technological strategies that allow better incorporation of these ingredients, not only to improve the nutritional profile but also to maintain the texture and enhance the sensory properties of the bread and consequently, increase the effects on consumer health.

Keywords: protein sources; nutritional improvement, texture, bread.

The following was modified in the introduction section:

Lines 41 - 47:

Enhancing nutrition in bread is an interesting opportunity for the food industry. Most of the bread are made with refined wheat flour because it generates a high loaf volume, light color, homogeneous crumb porosity and soft crumb. These bread are widely accepted by customers, although they lack vitamins, minerals, lysine, dietary fiber, antioxidant components and a high glycemic index [2,8,9]. This deficiency can be improved with whole wheat flour, but consumers prefer refined wheat bread [2] or other alternatives for better nutritional balance. 

Please also change the use of the term flour for powdered ingredients like vegetable and fruit waste, edible insects or animal waste - not all powder is flour. These ingredients are powder, not flour.

Author response:

Done. The changes are as follows:

Line: 53: powders

Line 78: flours or powders

Line 110: Powder protein sources

Line 164: cricket powder

Line 164: the powder

Line 215: cricket powder blends

Line 241: flour or powder

Tables 1, 2, 3 and 4. All tables and in the text, it was referred to as powder where applicable.

The reason why bread made from so-called white wheat flour is not nutritious is not because of the wheat but because of the use of highly refined flour. A simple solution to this problem is to use whole-wheat flour - the technological quality will deteriorate, but the nutritional quality will increase significantly - the same happens when protein additives are used.

Author response:

In this manuscript for nutritional improvement, whole wheat flour is mentioned as an alternative. However  the focus in this review is on adding different protein sources in refined wheat bread.

Lines 41 – 54:

Enhancing nutrition in bread is an interesting opportunity for the food industry. Most of the bread are made with refined wheat flour because it generates a high loaf volume, light color, homogeneous crumb porosity and soft crumb. These bread are widely accepted by customers, although they lack vitamins, minerals, lysine, dietary fiber, antioxidant components and a high glycemic index [2,8,9]. This deficiency can be improved with whole wheat flour, but consumers prefer refined wheat bread [2] or other alternatives for better nutritional balance.  The trend is to use alternative protein sources in refined wheat bread to enhance protein and fiber content and improve amino acid balance. Also, these ingredients could increase the antioxidant potential and reduce the glycemic index in bread [8,10]. Animal and vegetable ingredients such as legumes, cereals, pseudocereals, milk derivates, edible insects, fish derivates and other sources (Fig. 1) are commonly used in different ways, including flours, powders, protein concentrates, protein isolates and protein hydrolysates. Those protein sources can be added to bread by replacing refined wheat flour or additional ingredients.

Lines 63 – 67:

The objective of this review is to provide current information on the enrichment of refined wheat bread with protein ingredients from different sources, how it affects the dough behavior and technological properties of the bread matrix, as well as the impact of novel ingredients on sensory characteristics and the biologic potential of nutritionally improved bread.

So the authors should change the main context of the narrative and justify the advantage of using protein additives over making bread from whole grain flour, currently, the narrative justifying the use of additives is laconic and naive.

Author response:

The focus of this review is on the addition of different protein sources in refined wheat bread.

Lines 41 – 47:

Enhancing nutrition in bread is an interesting opportunity for the food industry. Most of the bread are made with refined wheat flour because it generates a high loaf volume, light color, homogeneous crumb porosity and soft crumb. These bread are widely accepted by customers, although they lack vitamins, minerals, lysine, dietary fiber, antioxidant components and a high glycemic index [2,8,9]. This deficiency can be improved with whole wheat flour, but consumers prefer refined wheat bread [2] or other alternatives for better nutritional balance.

Lines 53 - 62

These protein sources could improve the nutritional content and bring health benefits. However, other parameters such as dough rheology, texture, and other sensory characteristics, including appearance, flavor and taste are modified compared with traditional bread. The lack of gluten in the different protein sources interferes with the final product quality. Some authors prefer to use high protein ingredients (80-90%) in small amounts compared to whole flour, reaching the same protein content and lowering the wheat flour reduction [11].

Specific comments:

Line 48 - easy to obtain using wholegrain flour;

Author response:

It was changed:

Lines 41 – 47:

Enhancing nutrition in bread is an interesting opportunity for the food industry. Most of the bread are made with refined wheat flour because it generates a high loaf volume, light color, homogeneous crumb porosity and soft crumb. These bread are widely accepted by customers, although they lack vitamins, minerals, lysine, dietary fiber, antioxidant components and a high glycemic index [2,8,9]. This deficiency can be improved with whole wheat flour, but consumers prefer refined wheat bread [2] or other alternatives for better nutritional balance.

Line 64-66 - the bakery or only bread industry?, this line is a good example of non-justified generalisation; moreover in fact bakery industry is focused on delivering a lot of innovative and healthier products - so this line lacks sense because is too generalizing

Author response:

The word “bakery” was replaced by “bread”.

Lines 61 - 62: The bread industry should focus its attention on offering not only healthier products with high protein content, but also products with sensory characteristics attractive to consumers.

The word “bakery” was changed to” bread”, and “refined wheat bread”

Line 2: In title: Review Protein ingredients in bread: technological, textural and health implications

Line 61: bread industry

Line 86: bread products

Line 95: in bread

Line 253: bread product

Line 343: bread products

Line 389: bread products

Line 74 - healthier alternatives? Bread made from wholemeal flour is super healthy; you need to change the narrative in the whole article.

Author response:

The narrative was focused on the term “refined wheat bread”

Lines 63 – 67:

The objective of this review is to provide current information on the enrichment of refined wheat bread with protein ingredients from different sources, how it affects the dough behavior and technological properties of the bread matrix, as well as the impact of novel ingredients on sensory characteristics and the biologic potential of nutritionally improved bread.

Line 86 - this sentence is obvious; rewrite.

Author response:

It was eliminated

Line 103-104 - cite a reference for edible insects versus meat-nutritional comparison.

Author response:

Done:

The references 17 and 18 were added.

Line 100 – 101

The amino acid profile in insects is comparable to the meat with lower environmental affection [17,18]. Depending on the insect species used, the composition of protein, fat, micronutrients and fiber will differ [17,19,20]

References:

  1. Orkusz, A. Edible Insects versus Meat Nutritional Comparison: Knowledge of Their Composition Is the Key to Good Health. Nutrients 2021, 13, 1207, doi:10.3390/nu13041207.

  1. Gravel, A.; Doyen, A. The Use of Edible Insect Proteins in Food: Challenges and Issues Related to Their Functional Properties. Innov. Food Sci. Emerg. Technol. 2020, 59, 102272, doi:10.1016/j.ifset.2019.102272.

Line 106 - 108 - check the grammar of the sentence

Author response:

The term “source of protein” is quoted as is in the Regulation or the European Parliament.. The reference was added.

Lines 103 – 105:

European regulation indicates a claim that a food is a "source of protein" when it contains at least 12% of the energy value is provided by protein and a claim "high protein" when a food contains at least 20% of the energy value is provided by protein [21].

Reference:

Regulation (EC) No 1924/2006 of the European Parliament and of the Council of 20 December 2006 on nutrition and health claims made on foods Official Journal of the European Union Available online: https://eur-lex.europa.eu/legal-content/EN/TXT/PDF/?uri=CELEX:32006R1924&from=en (accessed on 28 July 2022).

Table 1 - for Lupin - should be debittered, not debittering

Author response:

Done.

In Table 1. Debittered Flour.

Table 1 and all tables - remove flour when not obtained from starchy grains - use the term powder instead

Author response:

Done. The changes are the next:

Line: 53: powders

Line 78: flours or powders

Line 110: Powder protein sources

Line 164: cricket powder

Line 164: the powder

Line 215: cricket powder blends

Line 241: flour or powder

Tables 1, 2, 3 and 4. All tables and in the text, it was referred to as powder where applicable.

Line 312-313 - the darker colour of the bread is not only the result of Maillard but also the innate colour of ingredients contributes to the final score.

Author response:

Done

Lines: 319- 323:

The appearance in terms of color formation in the "golden yellow" or brown crust of bread is called "browning”. It results from two different chemical reactions: Maillard reaction and caramelization during the baking process. The final color may vary depending on the bread product, raw materials and their processing, protein source, and the baking process conditions (cooking time and temperature) [20,61].

Reviewer 3 Report

This review summarized information on effects of added proteins from various sources on bread quality. This is a hot topic. The information is important.

My only concern is about Table 1. It seems obvious to me that adding higher percentage of proteins will change the composition of the resulting flour. I suggest authors simplify or delete this table. 

Normally, adding other proteins will dilute gluten and therefore negatively affects dough and bread quality. But some papers author cite showed some different results. Authors are suggested to highlight these novel findings.

Author Response

Reviewer 3

This review summarized information on effects of added proteins from various sources on bread quality. This is a hot topic. The information is important.

My only concern is about Table 1. It seems obvious to me that adding higher percentage of proteins will change the composition of the resulting flour. I suggest authors simplify or delete this table.

Author response:

Thank you very much for your comments.

The information in Table 1 shows different protein additions because the reader can compare it with the information in Tables 2, 3 or 4 if necessary.

Normally, adding other proteins will dilute gluten and therefore negatively affects dough and bread quality. But some papers author cite showed some different results. Authors are suggested to highlight these novel findings.

Author response:

Done.

The information was modified, highlighting positive results:

Lines 289 - 295:

Previous acidification of lupin flour by lactic fermentation changed the functional properties by increasing water absorption when the dough was integrated, reinforcing the gluten network and increasing the extension of resistance during fermentation generating a softer bread [44]. The integration of insects in bread can change the texture parameters, such as hardness and elasticity, due to the amount of insect fat that plasticizes and lubricates the dough, increasing the incorporation of air during kneading [20,33].

The conclusions were modified:

Lines 395 – 417:

The selection of protein sources, the amount of addition and the dough processing and baking methods are factors to consider when producing protein-rich bread. The addition of different protein sources influences from competition for water with gluten to form the dough network structure, due to competition for water with gluten to form, which affects carbon dioxide retention during fermentation. It can also impact crumb characteristics and sponginess during baking, which are associated with the texture and appearance of the bread.

Depending on the protein source, it could interact with the other ingredients, impacting the color of the bread crust and the taste of bread, influencing consumer acceptance. Therefore, processing strategies such as sourdough formation may favor the development of bread with improved textural properties, flavor, and protein digestibility benefits.

On the other hand, incorporating protein sources with functional properties in bread could provide health benefits, such as antioxidant potential, lower glycemic index, and enrichment in micronutrients such as vitamins and minerals. The lowering of glycemic index is related to the resistance of starch hydrolysis by digestive enzymes in response to different interactions between lipids, proteins and starch and the effect of polyphenols among digestive enzymes. Protein digestibility parameter change among treatments, where sourdough could be an excellent alternative to inactivate antinutritional factors.

Having technofunctional, sensory, and functional characteristics in the bread product is a great challenge for the food industry. Therefore, it is necessary to search for new strategies to incorporate different protein sources not only to improve the nutritional profile of bread but also to have a product with high quality characteristics, including good  texture profile and taste that could provide health benefit to consumers. 
